

# Computational analysis of microRNA-mediated interactions in SARS-CoV-2 infection

Müşerref Duygu Saçar Demirci[1] and Aysun Adan[2]

[1] Bioinformatics, Abdullah Gul University, Kayseri, Turkey
[2] Molecular Biology and Genetics, Abdullah Gul University, Kayseri, Turkey

## ABSTRACT

MicroRNAs (miRNAs) are post-transcriptional regulators of gene expression found in more than 200 diverse organisms. Although it is still not fully established if RNA viruses could generate miRNAs, there are examples of miRNA like sequences from RNA viruses with regulatory functions. In the case of Severe Acute Respiratory Syndrome Coronavirus 2 (SARS-CoV-2), there are several mechanisms that would make miRNAs impact the virus, like interfering with viral replication, translation and even modulating the host expression. In this study, we performed a machine learning based miRNA prediction analysis for the SARS-CoV-2 genome to identify miRNA-like hairpins and searched for potential miRNA-based interactions between the viral miRNAs and human genes and human miRNAs and viral genes. Overall, 950 hairpin structured sequences were extracted from the virus genome and based on the prediction results, 29 of them could be precursor miRNAs. Targeting analysis showed that 30 viral mature miRNA-like sequences could target 1,367 different human genes. PANTHER gene function analysis results indicated that viral derived miRNA candidates could target various human genes involved in crucial cellular processes including transcription, metabolism, defense system and several signaling pathways such as Wnt and EGFR signalings. Protein class-based grouping of targeted human genes showed that host transcription might be one of the main targets of the virus since 96 genes involved in transcriptional processes were potential targets of predicted viral miRNAs. For instance, basal transcription machinery elements including several components of human mediator complex (MED1, MED9, MED12L, MED19), basal transcription factors such as TAF4, TAF5, TAF7L and site-specific transcription factors such as STAT1 were found to be targeted.
In addition, many known human miRNAs appeared to be able to target viral genes involved in viral life cycle such as S, M, N, E proteins and ORF1ab, ORF3a, ORF8, ORF7a and ORF10. Considering the fact that miRNA-based therapies have been paid attention, based on the findings of this study, comprehending mode of actions of miRNAs and their possible roles during SARS-CoV-2 infections could create new opportunities for the development and improvement of new therapeutics.

Corresponding author
Müşerref Duygu Saçar Demirci,
duygu.sacar@agu.edu.tr

## INTRODUCTION

Coronoviruses (CoVs) are pathogens with serious health effects including enteric, respiratory, hepatic and central nervous diseases on human and animals. Zoonotic CoVs, Severe Acute Respiratory Syndrome-Coronavirus (SARS-CoV) and Middle East Respiratory Syndrome coronavirus, have been identified the sources of outbreaks in 2002/2003 and 2012, respectively with high mortality rates due to severe respiratory syndrome (*De Wilde et al., 2018*; *Chang, Yan & Wang, 2020*). Besides zoonotic CoVs, there are four types of human CoVs have been identified known as HCoV-OC43, HCoV-2293, HCoV-NL63 and HCoV-HKU1 (*De Wilde et al., 2018*). Unknown pneumonia have been detected in Wuhan, China and spread globally since December 2019. The World Health Organization named this new coronavirus as Severe Acute Respiratory Syndrome Coronavirus 2 (SARS-CoV-2) responsible for the new disease termed Coronavirus Disease 2019 (COVID-19) and it is the seventh identified CoV with animal origin infecting human (*Chang, Yan & Wang, 2020*).

SARS-CoV-2, a positive-single stranded RNA (+ssRNA) virus with exceptionally large genome with 5′cap structure and 3′polyA tail, belongs to β CoV with 45-90% genetic similarity to SARS-CoV based on sequence analysis and might share similar viral genomic and transcriptomic complexity (*Deng & Peng, 2020*; *Chen, Liu & Guo, 2020*). SARS-CoV-2 encodes nonstructural proteins while others code for structural proteins required for viral replication and pathogenesis. Structural proteins include spike (S) glycoprotein, matrix (M) protein, small envelope (E) protein and nucleocapsid (N) protein with various roles for virus entrance and spread (*Cheng & Shan, 2020*).

Currently, it has been also revealed that SARS-COV-2 has a very high homology with bat CoVs, which indicated how it is transmitted to human without knowing intermediate carriers (*Zhou et al., 2020*). The S protein of SARS-CoV-2 has a strong interaction with human angiotensin-converting enzyme 2 (ACE2) expressed on alveolar epithelial cells, which shows the way of virus infection in human (*Xu et al., 2020*).

MicroRNAs (miRNAs) are small, noncoding RNAs that play role in regulation of the gene expression in various organisms ranging from viruses to higher eukaryotes. It has been estimated that miRNAs might influence around 60% of mammalian genes and their main effect is on regulatory pathways including cancer, apoptosis, metabolism and development (*Li & Zou, 2019*).

Although the current release of miRNAs, the standard miRNA depository, lists miRNAs of 271 organisms, only 34 of them are viruses (*Griffiths-Jones et al., 2008*). While, the first virus-encoded miRNAs was discovered for the human Epstein–Barr virus (EBV) (*Pfeffer et al., 2004*), more than 320 viral miRNA precursors were reported so far. Although it has been shown that various DNA viruses express miRNAs, it is still debatable if RNA viruses could also encode. The major concerns regarding miRNAs of RNA viruses are based on (*Saçar Demirci, Toprak & Allmer, 2016*):

- the fact that RNA viruses that replicate in cytoplasm do not have access to nuclear miRNA machinery;
- the fact that since RNA is the genetic material, miRNA production would interfere with viral replication.
The involvement of both host miRNAs and viral miRNAs in viral infections have been discussed extensively although their exact mechanistic roles in disease pathogenesis are not fully understood. In general, host miRNAs are produced as a part of antiviral response at early stage of viral infections to cope with the infection by directly or indirectly targeting viral replication, transcription and translation. However, some viruses have the ability to manipulate this response to escape from the host defense system and induce their own replication by triggering degradation of host miRNAs or inhibiting their maturation (*Bruscella et al., 2017*). On the other hand, viral miRNAs are produced to regulate their own lytic or latency phase transitions and to regulate the expression of host mRNAs involved in antiviral responses or cell metabolism (*Bruscella et al., 2017*).

Currently, options for the prevention and treatment of CoVs are very limited due to the complexity. Therefore, detailed analysis of CoV-host interactions is quite important to understand viral pathogenesis and to determine the outcomes of infection. Although there are studies regarding to the viral replication and their interaction with host innate immune system, the role of miRNA-mediated RNA-silencing in SARS-CoV-2 infection has not been enlightened yet. In this study, SARS-CoV-2 genome was searched for miRNA-like sequences and potential host-virus interactions based on miRNA actions were analyzed.

## MATERIALS AND METHODS

Data analysis, pre-miRNA prediction, mature miRNA detection workflows were generated by using the Konstanz Information Miner (KNIME) platform (*Berthold et al., 2008*)

### Data

Genome data of the virus was obtained from NCBI: Severe acute respiratory syndrome coronavirus 2 isolate Wuhan-Hu-1, complete genome GenBank: MN908947.3.

MiRNA prediction workflow izMiR (*Saçar Demirci, Baumbach & Allmer, 2017*) and its related data were taken from Mendeley Data: https://data.mendeley.com/datasets/mgh5r9wny7/1.

Mature miRNA sequences of human were downloaded from miRBase Release 22.1 (*Kozomara & Griffiths-Jones, 2011*).

### Pre-miRNA prediction

Genome sequence of SARS-CoV-2 were transcribed (T→U) and divided into 500 nt long fragments with 250 overlaps. Then these fragments were folded into their secondary structures by using RNAfold (*Hofacker, 2003*) with default settings and hairpin structures were extracted, producing 950 hairpins in total.

A modified version of izMiR (SVM classifier is changed to Random Forest and latest miRBase version was used for learning) was applied to these hairpins with ranging lengths (from 7 to 176) (Fig. 1). Firstly, 1,917 human precursor miRNAs from miRBase were used as positive data and izMiR training workflow was applied with 70% learning—30% testing ratios, 1,000 times. Since izMiR method is a consensus approach, at the end of learning phase, 39,000 models were created for 13 feature groups and 3 classifiers (Decision Tree, Naive Bayes and Random Forest). By selecting models with the highest

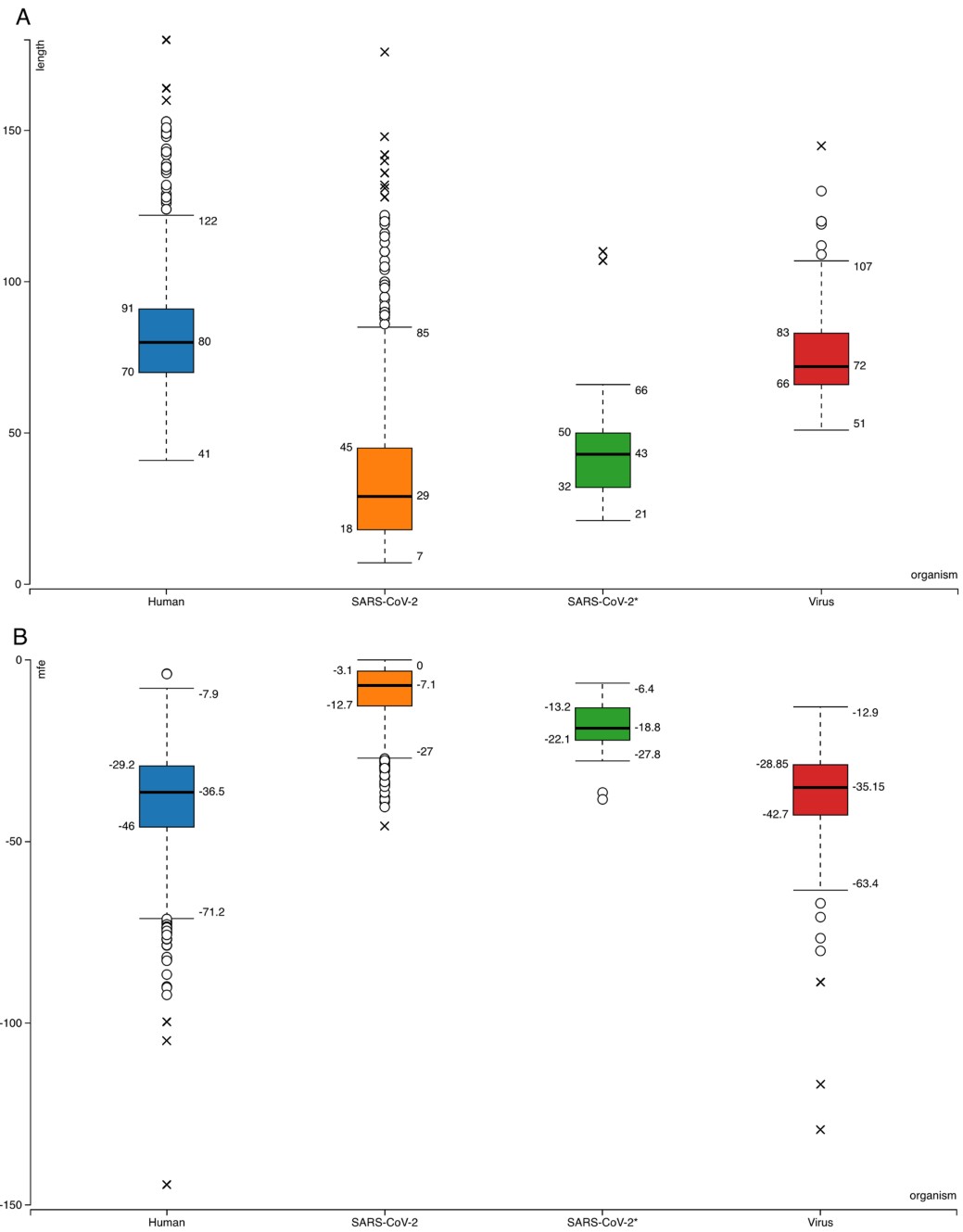

**Figure 1 Box-plots for comparison of general features of human pre-miRNAs and virus pre-miRNAs from miRBase and SARS-CoV-2 hairpins.** (A) Hairpin lengths. (B) Minimum free energy scores obtained from RNAfold (*Hofacker, 2003*). SARS-CoV-2 refers to all 950 hairpins extracted from the virus genome while SARS-CoV-2* indicates 29 selected hairpins.

accuracy scores, 39 models were applied to 950 viral hairpins. In order to select candidate pre-miRNA like sequences for further analysis, average of each classifiers' prediction scores was calculated. Then, overall average prediction score was calculated for each hairpin and by using minimum 0.900 as a threshold value, 29 hairpins were used for further analysis.

The same learning workflow was used to create models by using 320 virus hairpin sequences from miRBase as positive data instead of human. None of the 950 viral hairpins had minimum 0.900 overall average prediction score from these models' predictions (Supplemental File—viral hairpins).

Based on the mean value of averages of three classifiers' prediction scores (Decision Tree, Naive Bayes and Random Forest), 29 hairpins passed 0.900 threshold and used for further analysis.

## Mature miRNA prediction and targeting

Selected hairpins were further processed into smaller sequences; maximum 23 nt length with 6 nt overlaps. Then, these fragments were filtered based on minimum length of 15 and their location on the hairpins (sequences not involving any loop nucleotides were included). Target search of these remaining 30 candidate mature miRNAs were performed against human and SARS-CoV-2 genes by using psRNATarget tool with default settings (*Dai & Zhao, 2011*). Moreover, human mature miRNAs' from miRBase were applied for searching their targets in SARS-CoV-2 genes.

## Gene ontology

The targets of viral miRNAs in human genes were further analyzed for their Gene Ontology (GO). To achieve this, PANTHER Classification System (http://www.pantherdb.org) was used (*Mi, Muruganujan & Thomas, 2013*).

# RESULTS

## Pre-miRNA like sequence prediction from SARS-CoV-2 genome

Searching SARS-CoV-2 genome for sequences forming hairpin structures resulted in 950 hairpins with varying lengths (Supplemental Files—viral hairpins). In order to use machine learning based miRNA prediction approach of izMiR workflows, hundreds of features were calculated for all of the pre-miRNA candidate sequences. Among those, minimum free energy (Mfe) values required for the folding of secondary structures of hairpin sequences and hairpin sequence lengths of known human and virus miRNAs from miRBase and predicted hairpins of SARS-CoV-2 were compared (Fig. 1). Based on the box-plots shown in Fig. 1, most of the extracted viral hairpins seemed to be smaller than human miRNA precursors. Known viral and human pre-miRNAs share similarities both in length and mfe ranges. This could be due to the fact that viruses would need to use at least some members of host miRNA biogenesis pathway elements, therefore, viral miRNAs should be similar to host miRNAs to a certain degree. Therefore, a classification scheme trained with known human miRNAs was applied on SARS-CoV-2 hairpins. Only 29 hairpins out of 950 passed the 0.900 prediction score threshold (Fig. 1—SARS-CoV-2*) and used for further analysis. From these hairpins, 30 mature miRNA candidates were extracted and their possible targets for human and SARS-CoV-2 genes were investigated.

The same classification approach was used to build models based on known viral hairpins too. However, these models' predictions on SARS-CoV-2 hairpins did not provide

any candidates with average prediction scores equal or above 0.900 (Supplemental File—viral hairpins). This could be due to limited number of positive miRNA samples (320 hairpins for viruses) used for learning and/or quality of the datasets which is one of the most important elements of a classification analysis (*Demirci & Allmer, 2017*).

## Sequence similarity

SARS-CoV-2 miRNA candidates were further analyzed to test if they were similar to any of the known mature miRNAs from 271 organism listed in miRBase. To achieve this, a basic similarity search was performed based on the Levenshtein distance calculations in KNIME. However, there was no significant similarity between hairpin or mature sequences.

## Human gene targets of viral miRNA-like RNAs

Predicted mature miRNA-like sequences of SARS-CoV-2 were used to find their targets in SARS-CoV-2 and human genes, mature miRNAs of human were also applied on SARS-CoV-2 genes (Table 1; Supplemental Files—virus to human targets, virus to virus targets). Although miRNA based self-regulation of viral gene expression is a hypothetical case, SARS-CoV-2 ORF1ab polyprotein gene might be the only one that could be a target of viral miRNAs. In total 1,367 human genes seemed to be targeted by viral miRNAs.

It has been shown that viruses could lower host transcription to benefit viral gene expression and also reduce the effect of the immune system (*Lyles, 2000*). "Host shutoff" is a phenomenon observed when several human viruses including SARS, lead to a global decrease in the host protein production (*Harwig, Landick & Berkhout, 2017*). Since host transcription process appear to be an important target for viruses, Table 1 lists some of the predicted targets of SARS-CoV-2 miRNAs in human genes that have roles in transcription. The full list of miRNA—target predictions are available in Supplemental Files.

## SARS-CoV-2 gene targets of human miRNAs

Considering the potential of miRNAs as biomarkers and therapeutic agents, it is essential to check if any of the known human miRNAs could target viral genes. Among 2,654 mature entries of *Homo sapiens* in miRBase Release 22.1, 479 of them could target SARS-CoV-2 genes (Supplemental Files—human to virus). While Envelope and ORF6 genes were targeted by single miRNAs, ORF1ab appeared to be the target of 369 different human mature miRNAs (Table 2). As expected, number of targeting events appeared to be correlated with the gene length.

## GO of targeted human genes

Lastly, in order to understand the main mechanisms that would be affected by the influence of viral miRNA-like sequences on human genes, PANTHER Classification System was applied to targeted human genes. Based on the results presented in Figs. 2 and 3 and Supplemental Files, a wide range of human genes with various molecular functions and pathways could be targeted.

**Table 1 Transcription related human gene targets of viral miRNAs.** MiRNA column indicates the sequences of mature viral miRNAs; Target gene # column shows the total number of different genes involved in transcription and targeted by the corresponding viral miRNAs.

| miRNA | Target gene # | Target gene names |
|---|---|---|
| GUUUUCAUCAACUUUUAAC | 11 | CNOT4 (CCR4-NOT transcription complex subunit 4), MED9 (Mediator of RNA polymerase II transcription subunit 9), GTF2H5 (General transcription factor IIH subunit 5), MED1 (Mediator of RNA polymerase II transcription subunit 1), STAT5B (Signal transducer and activator of transcription 5B), TAF4 (Transcription initiation factor TFIID subunit 4), EBF1 (Transcription factor COE1), CNOT10 (CCR4-NOT transcription complex subunit 10), MAFG (Transcription factor MafG), BACH1 (Transcription regulator protein BACH1), MED12L (Mediator of RNA polymerase II transcription subunit 12-like protein) |
| ACGUUGCAAUUUAGGUGGUGC | 4 | CNOT4 (CCR4-NOT transcription complex subunit 4), TFDP2 (Transcription factor Dp-2), TCF4 (Transcription factor 4), MITF (Microphthalmia-associated transcription factor) |
| AGCUAGCUCUUGGAGGUUCCGUG | 3 | LST1 (Leukocyte-specific transcript 1 protein), EBF4 (Transcription factor COE4), TFEC (Transcription factor EC) |
| AUAAGCUCAUGGGACACUUCGCA | 3 | HES2 (Transcription factor HES-2), TAF5 (Transcription initiation factor TFIID subunit 5), TFEC (Transcription factor EC) |
| UAUGUACCACUAAAGUCUGCUAC | 3 | SOX11 (Transcription factor SOX-11), MED19 (Mediator of RNA polymerase II transcription subunit 19), NFYB (Nuclear transcription factor Y subunit beta) |
| UUGAUAAAGUACUUAAUGAGAAG | 2 | TEAD1 (Transcriptional enhancer factor TEF-1), DMRT1 (Doublesex- and mab-3-related transcription factor 1) |
| AAGUACUUAAUGAGAAGUGCUCU | 2 | TFDP2 (Transcription factor Dp-2), TCF4 (Transcription factor 4) |
| AUUUAGGUGGUGCUGUCUGU | 2 | CTCFL (Transcriptional repressor CTCFL), CNOT6L (CCR4-NOT transcription complex subunit 6-like) |
| CAUGUAUUCUGUUAUGCUUACUA | 2 | TRPS1 (Zinc finger transcription factor Trps1), CREBZF (CREB/ATF bZIP transcription factor) |
| CUGCCUAUACAGUUGAACUCGGU | 1 | BRF1 (Transcription factor IIIB 90 kDa subunit) |
| GUACCACUAAAGUCUGCUACGUG | 1 | NFYB (Nuclear transcription factor Y subunit beta) |
| AACAAAGCUAGCUCUUGGAGGU | 1 | SUPT5H (Transcription elongation factor SPT5) |
| UCCGUGGCUAUAAAGAUAACAGA | 1 | MYT1L (Myelin transcription factor 1-like protein) |
| UCAUGGGACACUUCGCAUGGUGG | 1 | PHTF2 (Putative homeodomain transcription factor 2) |
| CCUGUGUUGUGGCAGAUGCUGUC | 1 | TAF7L (Transcription initiation factor TFIID subunit 7-like) |
| UUGUGGCAGAUGCUGUCAUAAAA | 1 | POU2F1 (POU domain, class 2, transcription factor 1) |
| AUAGAUUAUGUACCACUAAAGUC | 1 | STAT1 (Signal transducer and activator of transcription 1-alpha/beta) |
| CAACCUAUACUGUUACUAGAUCA | 1 | SWT1 (Transcriptional protein SWT1) |

According to the protein classes of targeted human genes, metabolite interconversion enzyme (110 genes) and gene-specific transcriptional regulator (96 genes) were the most likely candidates as targets of viral miRNA-like sequences. For the defense/immunity protein class, six genes (IGSF1, IGSF3, VSTM4, CD48, CD84, PI15) appeared to be targeted.

Pathway based GO analysis revealed that overall 101 pathways might be influenced by viral miRNA like sequences. Among them, Gonadotropin-releasing hormone receptor pathway was the most targeted with 25 genes, followed by various signaling pathways including Wnt signaling pathway (21 genes), EGF receptor signaling pathway (17 genes), CCKR signaling map (16 genes), FGF signaling pathway (16 genes) and PDGF signaling pathway (15 genes) (Fig. 3).

**Table 2 Predicted viral mRNA targets by human miRNAs: bold miRNAs are the common ones targeting more than one indicated viral proteins.** The functions of SARS-CoV-2 proteins are not fully characterized, however, its coding genes might share functional similarity with SARS-CoV as shown in column "Functions of Target Genes". Gene size indicates the size of genes in terms of number of nucleotides, hsa miRNAs shows the number of different human miRNAs that could target indicated viral genes.

| Target genes | Human miRNA | Functions of target genes |
|---|---|---|
| **S (Spike) protein** <br> gene size: 3,822 <br> hsa miRNAs: 67 | hsa-miR-447b, **hsa-miR-2052, hsa-miR-3127-5p, hsa-miR-34b-5p, hsa-miR-374a-3p, hsa-miR-6729-5p, hsa-miR-3927-3p, hsa-miR-410-5p, hsa-miR-432-5p, hsa-miR-4693-3p, hsa-miR-548ag, hsa-miR-6128, hsa-miR-676-3p, hsa-miR-6809-5p, hsa-miR-6893-5p** | Viral attachment for the host cell entry by interacting with ACE2 (*Gallagher & Buchmeier, 2001*; *Xu et al., 2020*) |
| **E (Envelope) protein** <br> gene size: 228 <br> hsa miRNAs: 1 | **hsa-miR-3672** | Viral envelope formation and acting as viroporin to form hydrophilic pores on host membranes (*Wilson et al., 2004*; *Castaño-Rodriguez et al., 2018*) |
| **M (Membrane) protein** <br> gene size: 669 <br> hsa miRNAs: 10 | hsa-miR-325, hsa-miR-34a-5p, **hsa-miR-6820-5p**, hsa-miR-1252-5p, hsa-miR-1262, hsa-miR-2355-3p, hsa-miR-382-5p, hsa-miR-215-3p, **hsa-miR-5047**, hsa-miR-6779-5p | Defining the shape of viral envelope, the central organizer of CoV assembly (*Masters, 2006*; *Neuman et al., 2011*) |
| **N (Nucleocapsid) protein** <br> gene size: 1,260 <br> hsa miRNAs: 21 | hsa-miR-8066, hsa-miR-1911-3p, hsa-miR-4259, hsa-miR-6838-3p, hsa-miR-208a-5p, hsa-miR-4445-5p, hsa-miR-451b, hsa-miR-6082, hsa-miR-8086, hsa-miR-1282, hsa-miR-1301-3p, hsa-miR-154-5p, **hsa-miR-1910-3p**, hsa-miR-3155a, **hsa-miR-342-5p, hsa-miR-593-3p**, hsa-miR-639, **hsa-miR-6729-5p**, hsa-miR-6741-5p, hsa-miR-6876-5p, hsa-miR-6882-3p | Only protein primarily binding to the CoV RNA genome to form nucleocapsid (*Masters, 2006*) |
| **ORF1ab** <br> gene size: 21,291 <br> hsa miRNAs: 369 | hsa-miR-153-5p, hsa-let-7c-5p, **hsa-miR-1910-3p, hsa-miR-342-5p, hsa-miR-4436b-3p, hsa-miR-5047, hsa-miR-203b-3p, hsa-miR-2052, hsa-miR-3127-5p, hsa-miR-3190-3p, miR-34b-5p, hsa-miR-3672, hsa-miR-374a-3p, hsa-miR-3927-3p, hsa-miR-410-5p, hsa-miR-432-5p, hsa-miR-4436a, hsa-miR-4482-3p, hsa-miR-4693-3p, hsa-miR-5011-3p, hsa-miR-548ag, hsa-miR-593-3p, hsa-miR-6128, hsa-miR-676-3p, hsa-miR-6809-5p, hsa-miR-6820-5p, hsa-miR-6866-5p, hsa-miR-6893-5p** | Encoding 5′- viral replicase (*Graham et al., 2008*) |
| **ORF3a** <br> gene size: 828 <br> hsa miRNAs: 16 | hsa-miR-549a-3p, hsa-miR-1246, hsa-miR-7704, **hsa-miR-203b-3p, hsa-miR-342-5p**, hsa-miR-4422, hsa-miR-4510, hsa-miR-1229-5p, hsa-miR-190b-5p, hsa-miR-203a-3p, hsa-miR-367-5p, **hsa-miR-4436b-3p**, hsa-miR-4482-3p, hsa-miR-541-3p, hsa-miR-6751-5p, hsa-miR-6891-5p, **hsa-miR-4482-3p** | a sodium or calcium ion channel protein, involved in replication and pathogenesis together with E and ORF8a (*Castaño-Rodriguez et al., 2018*) |
| **ORF8** <br> gene size: 366 <br> hsa miRNAs: 13 | hsa-miR-12129, **hsa-miR-5047**, hsa-miR-148a-3p, hsa-miR-23b-5p, **hsa-miR-5011-3p**, hsa-miR-12119, hsa-miR-2392, **hsa-miR-3190-3p**, hsa-miR-3529-5p, hsa-miR-369-3p, hsa-miR-455-5p, hsa-miR-4779, hsa-miR-648 | Might be important for interspecies transmission (*Lau et al., 2015*; *Castaño-Rodriguez et al., 2018*) in addition to its roles in replication |
| **ORF7a** <br> gene size: 366 <br> hsa miRNAs: 8 | **hsa-miR-4436a**, hsa-miR-3135b, **hsa-miR-4436b-3p**, hsa-miR-4774-5p, hsa-miR-6731-5p, **hsa-miR-6866-5p, hsa-miR-1910-3p**, hsa-miR-5590-3p | accessory protein that is composed of a type I transmembrane protein, induction of apoptosis in a caspase-dependent pathway (*Tan et al., 2007*; *Schaecher et al., 2007*) |
| **ORF10** <br> gene size: 117 <br> hsa miRNAs: 4 | hsa-miR-3682-5p, hsa-miR-411-5p, hsa-miR-379-5p, hsa-miR-548v | Might be involved in transspecies transmission (*Paraskevis et al., 2020*) |
| **ORF6** <br> gene size: 186 <br> hsa miRNAs: 1 | hsa-miR-190a-5p | Blocking nuclear import of STAT1 by binding to nuclear imports (*Frieman et al., 2007*) |
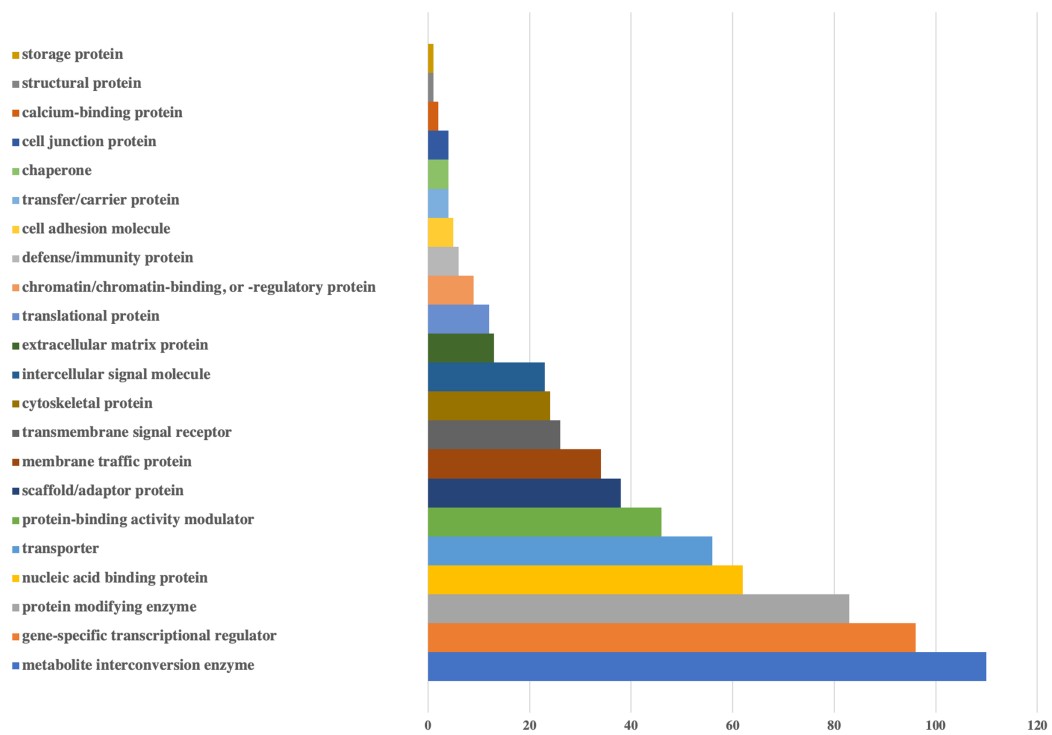

**Figure 2 Bar-chart for the protein classes of human genes that could be targeted by viral miRNAs.** Protein classes of genes were obtained from Panther. *X*-axis shows the number of genes with respected classes.

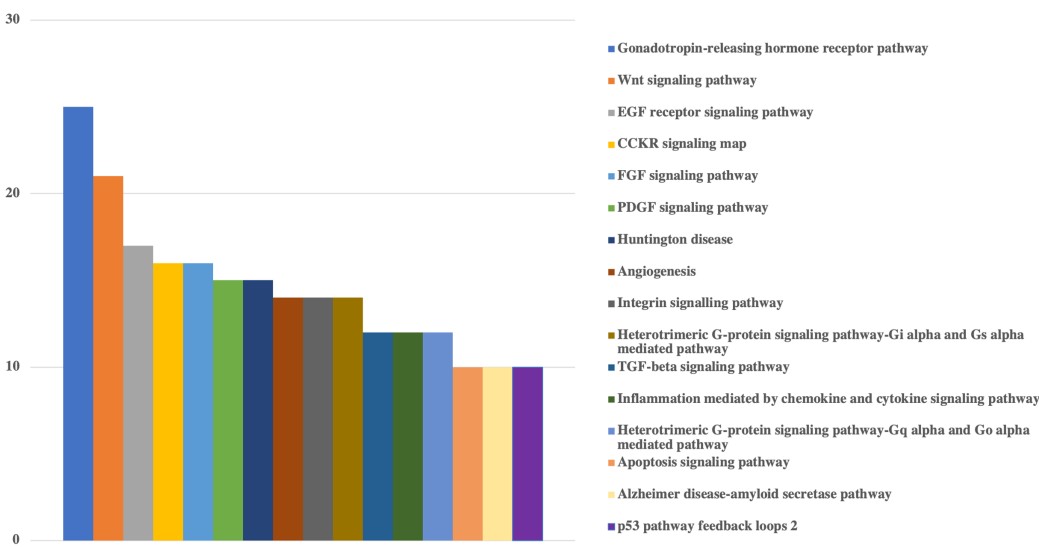

**Figure 3 Bar-chart for the pathways of human genes that could be targeted by viral miRNAs.** Graph is limited to the pathways that have at least 10 genes. Pathways of genes were obtained from Panther. *Y*-axis shows the number of genes with respected pathways. Chart and legend are sorted from maximum to minimum (left to right and top to bottom, respectively).

By comparing the predicted human gene targets of SARS-CoV-2 miRNA candidates with recent protein–protein interactions (PPI) data between human and virus (*Gordon et al., 2020*), 28 common proteins were identified (Supplemental File—PPI). Thus, these human proteins might not only be associated with viral proteins but also be targets of viral miRNA-like sequences.

## DISCUSSION

The potential roles of miRNA-mediated RNA interference in infection biology has been defined as an essential regulatory molecular pathway. The involvement of miRNAs in host–pathogen interactions during infection might include the targeting of viral genes by host-cellular miRNAs as well as evolution of DNA and RNA virus-based gene silencing mechanisms to overcome host antiviral response and to maintain viral infection and disease (*Ghosh, Mallick & Chakrabarti, 2009*; *Piedade & Azevedo-Pereira, 2016*).

In addition, viruses might use host cellular miRNAs for their own advantage. Therefore, molecular elucidation of the roles of miRNAs in host–virus interaction might provide a deeper understanding for viral pathogenesis and the development of an effective antiviral therapy as shown in several viruses including Herpesvirus, Enterovirus and Hepatitis C (*Piedade & Azevedo-Pereira, 2016*; *Engelmann et al., 2018*; *Khatun & Ray, 2019*). Although the detailed knowledge about viral miRNAs have been obtained from DNA viruses, it is still controversial for RNA viruses that whether they produce their own miRNAs or not (*Mishra et al., 2019*). However, there are several reports showing the presence of non-canonical (due to the lack of classical stem-loop structure in miRNAs), small miRNA-like small RNAs produced during viral infections as shown in H5N1 Influenza (*Li et al., 2018*), Ebola virus (*Liang et al., 2014*) and HIV-1 (*Klase et al., 2009*). Although they play crucial roles in several viral infections, miRNAs have not been studied in the pathogenesis of SARS-CoV-2 infection. In this study, we elucidated the presence of miRNAs in SARS-CoV-2 infection by predicting possible host genes targeted by viral miRNA-like small RNAs and viral genes targeted by cellular miRNAs, which might provide potential ways to understand the underlying mechanisms of SARS-CoV-2 infection.

We investigated SARS-CoV-2 encoded genes targeted by host-cellular miRNAs as shown in Table 2, which are mainly responsible for viral biogenesis, entrance, replication and infection. Except envelope (E) protein and ORF6, all viral genes (S, M, N, ORF1ab, ORF3a, ORF8, ORF7a and ORF10) are targeted by multiple human miRNAs. For instance, hsa-miR-203b-3p targeted ORF1ab and ORF3a with roles in viral replication was already shown to suppress influenza A virus replication (*Zhang et al., 2018*). Even though hsa-miR-148a-3p targeted ORF8 to prevent interspecies transmission and also replication, it was found to target S, E, M and ORF1a protein in closely related SARS-CoV (*Mallick, Ghosh & Chakrabarti, 2009*). hsa-let-7c-5p targeted ORF1ab in SARS-CoV-2 while it was found to be involved in H1N1 influenza A suppression by targeting its M1 protein (*Ma et al., 2012*). In another study, ORF6 of SARS-CoV suppressed type I interferon signaling by blocking the nuclear transport of STAT1 in the presence of interferon β (*Huang et al., 2017*), therefore, hsa-miR-190a-5p might target ORF6 to

overcome immune system escape in SARS-CoV-2. The presence of such miRNAs could be considered as a host's innate antiviral defense mechanism. On the other hand, virus could use these miRNAs to suppress their own replication to escape from immune system at the beginning of infection and transmission for a stronger infection. For instance, miR-146a-5p was upregulated in hepatitis C virus-infected liver cells of patients and in infected human hepatocytes, which promoted virus particle assembly (*Bandiera et al., 2016*). Moreover, ss RNA viruses could evolve very rapidly to change their gene sequences matching with these host miRNAs, therefore, they increase their host specificity. Once the virus establishes a successful transmission inside the host, they would mutate their genes very rapidly to escape from host miRNAs, which results from their RNA polymerases without proofreading activity (*Ye, Montalto-Morrison & Masters, 2004*; *Trobaugh & Klimstra, 2017*).

Virus-derived miRNAs might function by targeting host and virus-encoded transcripts to regulate host–pathogen interaction. The roles of viral miRNAs in pathogenesis include alteration of host defense mechanisms and regulation of crucial biological processes including cell survival, proliferation, modulation of viral life-cycle phase (*Ahmad et al., 2020*). Although encoding miRNAs seems quite problematic for RNA viruses due to the nature of miRNA biogenesis pathway, it is possible to circumvent these problems through different ways as seen in HIV-1 (*Klase et al., 2009*). Therefore, we analyzed possible human genes targeted by predicted miRNA like small RNAs (Supplemental File) and focused on the genes involved in transcription (Table 1). Based on the panther analysis, regulators of eukaryotic transcription would be the most important targets of 18 SARS-CoV-2 derived mature miRNA like candidates out of 29 hairpins in total. These transcriptional regulators are involved in both basal transcription machinery including several components of human mediator complex (MED1, MED9, MED12L, MED19) and basal transcription factors such as TAF4, TAF5 and TAF7L. Viruses might downregulate host gene expression in order to increase their gene expression either co-transcriptionally in the nucleus or post-transcriptionally in the nucleus or cytoplasm (*Herbert & Nag, 2016*). Therefore, targeting basal transcription machinery such as components (TAFs) of TFIID complex by SARS-CoV-2 could prevent RNA polymerase II to assemble on promoters of host genes at the initiation step. Viral factors have been shown to block transcription initiation by inhibiting TFIID or more specifically TAF4 in herpesvirus (*Yang & Chang, 2013*). Another interesting target gene by SARS-CoV-2 miRNA was different subunits of CCR4-NOT transcription complex including CNOT4, CNOT10 and CNOT6L, which are deadenylases involved in mRNA decay (*Abernathy & Glaunsinger, 2015*). Therefore, suppression of these genes by viral miRNAs could impede mRNA turnover in the host and provide opportunities for viral mRNA to escape from degradation. Additionally, site specific trans-acting factors such as MAFG, STAT1, STAT5B and SOX11 would be targeted by viral miRNAs. STAT family transcription factors are activated by cytokine-induced stimuli and generally involved in an interferon response. In Kaposi's sarcoma-associated herpesvirus (KSHV), viral miRNAs were found to inhibit STAT3 and STAT5, resulting in deregulated interferon response and transition into lytic viral replication (*Ramalingam & Ziegelbauer, 2017*). In a recent study,

interactions between 26 SARS-COV-2 proteins and human proteins were investigated by using affinity-purification mass spectrometry (*Gordon et al., 2020*) and the results were found to be in accordance with some results presented in this study. When we compared our predicted SARS-COV2 miRNA like RNA targets in human with these 332 human proteins, we found out that there are 28 protein targets in common (Supplemental File—PPI) and these targets are involved in biological processes including protein trafficking, translation, transcription and ubiquitination regulation similar to our Panther analysis results.

## CONCLUSIONS

As a conclusion, viral diseases have been paid attention as a global health problem due to the lack of proper treatment strategies and rapid evolution of viruses. In recent years, studies have focused on identifying miRNAs as targets for the treatment of viral diseases and there are potential miRNA therapeutics under investigation which aim to overexpress or replace, inhibit or repress miRNAs in the cells or tissues Based on the results shown in Table 2, it can be concluded that increases in the level of host miRNAs targeting virulent genes such as S, M, N, E and ORF1ab would block viral entry and replication. Moreover, decreasing the levels of host miRNAs would make SARS-CoV-2 more replicative and visible for the host immune system. However, alterations in host miRNA levels would interfere with specific cellular processes which are crucial for the host biology. In our study, we have also identified possible miRNA like small RNAs from SARS-CoV-2 genome which target important human genes. Therefore, antagomirs targeting viral miRNAs could be also designed even though there are only a few studies for DNA viruses (*Herbert & Nag, 2016*). Antagomirs have been the most studied miRNA inhibiting approach among other therapeutics including small molecule inhibitors and miRNA sponges and there are promising clinical phase studies evaluating the therapeutic potential of antagomirs targeting host miRNAs such as miR-122 in hepatitis C (*Van Der Ree et al., 2017*).

On the other hand, studies have also continued to overcome some obstacles including difficult entrance into the target cells, in vivo instability and lower binding affinities to their targets in order to increase the effectiveness of antagomirs (*Simonson & Das, 2015*). In addition to inhibition of miRNAs in the target cells, some miRNAs with lower expression should be overexpressed or replaced by using miRNA mimics. Most miRNA mimics as with antagomirs need to be modified to prevent their degradation and increase their uptake by the cells (*McCaskill et al., 2017*). However, all these therapeutic possibilities need further mechanistical evaluations to understand how they regulate virus-host interaction. Therefore, further in vitro, ex vivo and in vivo studies will be required to validate candidate miRNAs for SARS-CoV-2 infection.

### Funding

The authors received no funding for this work.
## Competing Interests

The authors declare that they have no competing interests.

## Author Contributions

- Müşerref Duygu Saçar Demirci conceived and designed the experiments, performed the experiments, analyzed the data, prepared figures and/or tables, authored or reviewed drafts of the paper, and approved the final draft.
- Aysun Adan conceived and designed the experiments, prepared figures and/or tables, authored or reviewed drafts of the paper, and approved the final draft.

## Data Availability

Raw data is available as a Supplemental File.

## Supplemental Information

Supplemental information for this article can be found online at http://dx.doi.org/10.7717/peerj.9369#supplemental-information.

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
