# Peer review of "Computational analysis of microRNA-mediated interactions in SARS-CoV-2 infection"

_PeerJ, doi:10.7717/peerj.9369_

## Round 0.1 · original submission · Major Revisions

Please do address carefully all the reviewers' comments.

Reviewer 1 ·

Basic reporting

In this study, the authors have predicted through computational analyses several potential miRNAs, from SARS-COV2 that might play a role during infection.
A major concern relates to whether RNA viruses encode for miRNAs, which is a debated issue as also stated by the authors. This study should provide more support about this central point. For example, it would be useful to compare in Fig.1 the features for predicted SARS-COV2 miRNAs with those of known viral miRNAs from miRBase, in addition to the human ones. How the predictions would change by training the predictor using only features derived from the viral miRNAs ?
Some of the predicted miRNA should be experimentally validated.
Does any of the target gene of the predicted SARS-COV2 miRNAs participate to the recently characterized viral interactome (https://www.biorxiv.org/content/10.1101/2020.03.22.002386v1), either directly or indirectly by participating to the same biological processes/pathways?

Experimental design

no comment

Validity of the findings

no comment

·

Basic reporting

no comments

Experimental design

no comments

Validity of the findings

Manuscript bring a set of predictions on SARS-CoV2 and human miRNAs and their targets. It is a good set of results and contribute to a yet not discussed point in the COVID-19 pandemic, but need to be much more explored in terms of mechanisms, results description, possible networks identified by epigenetic regulation. Authors should also discuss limitations on the use of antagomIRs as therapeutic "drugs".

Additional comments

The manuscript from Demirci and Adan describes a computational analysis of miRNAs mediated networks in SARS-CoV2 infection. Authors seek for viral miRNAs and correlate them with human mRNA targets. Moreover authors do the opposite, looking for human miRNAs and their targets in viral genome.
It is a well written manuscript, with a good set of results that can only predict miRNAs interactions. It starts filling the gap in literatura about epigenetic regulation in COVID-19.
However there are some points:
1. Abstract: the abstract is well written but it does not bring to readers the main findings of the manuscript. There is an example of STAT1 signaling but no further conclusion is presented. I suggest authors rewrite the abstract with this issue in mind.
2. Line 41 – delete “there are”
3. Introduction: authors make an extensively description of the origins of SARS viruses and pandemic COVI-19. This makes the introduction too long. Moreover, authors do not explore the theme “miRNAs and virus infections”. Introduction should be re-written.
4. Methodology: it is not clear in this section, specifically “pre-miRNA prediction, the criteria used in the izMiR algorithm. Authors state they used 3 classifiers to generate scores and only 29 hairpins passed the threshold of 0.9. What does it mean? Which criteria? Why authors assume that virus miRNAs should have high similarity to human miRNAs?
5. Results:
5.1. 30 mature miRNAs were identified in the first analysis. It would be important to show the list of these miRNAs.
5.2. In lines 145-148 authors mention they were not able to identify similarities of the 30 selected mature viral miRNAs with human miRNAs. Maybe, it would be much more interesting if authors look for targets for these “possible viral miRNAs”. Maybe, there is no similarity with humans, but there is a correlation in sequence with mRNA targets. Indeed, authors say in line 154 that they were able to find about 1367 human targets for viral miRNAs. And they just show in table 1 some of them. What was the criteria to select only these targets? Why look at just for transcription regulators? How about control of immune response, cell signaling, etc.?
5.3. Lines 152-154 – authors say that SARS-CoV2 ORF 1ab is the only one that might be targeted by miRNAs. Why? Has it been described elsewhere? Which result demonstrate this?
5.4. Lines 165-168 – authors do not explore the results of figure 2 and 3. There is only a mention about them without describing the results properly.
6. The set of results presented in this manuscript represent a good analysis of viral and human miRNAs and their targets. However, authors describe the results in a confused way. It is strongly recommended that authors reorganize results section in order to stablish two main points: (i) viral miRNAs and their targets and (ii) human miRNAs and their targets.
7. Discussion:
7.1. Lines 188-189 - Change the sentence “in this study, we elucidated miRNA-mediated regulation responsible for SRS-CoV2 infection...” Results presented here are much more the characterization of miRNAs and their targets related to SARS-CoV2.
7.2. Line 191 – “…possible virulent genes…”What does virulent means?
8. Conclusion: authors should include limitation on the use o antagomIRs or miras agonists for therapeutics
Figure 2 and 3 are in low quality.
Table 2 title: predicted “viral” mRNA targets

---

## Round 0.2 · accepted · Accept

While the reviewers did not formally re-review the manuscript, Reviewer 1 has indicated that s/he is happy with the revisions.